# Features of Online Hospital Appointment Systems in Taiwan: A Nationwide Survey

**DOI:** 10.3390/ijerph16020171

**Published:** 2019-01-09

**Authors:** Po-Chin Yang, Feng-Yuan Chu, Hao-Yen Liu, Mei-Ju Shih, Tzeng-Ji Chen, Li-Fang Chou, Shinn-Jang Hwang

**Affiliations:** 1Department of Family Medicine, Taipei Veterans General Hospital, Taipei 112, Taiwan; michael00557@gmail.com (P.-C.Y.); steven2259898@gmail.com (F.-Y.C.); yen.ee93@gmail.com (H.-Y.L.); sjhwang@vghtpe.gov.tw (S.-J.H.); 2Graduate Institute of Communication Engineering, National Taiwan University, Taipei 112, Taiwan; b96901063@ntu.edu.tw; 3School of Medicine, National Yang-Ming University, Taipei 112, Taiwan; 4Department of Public Finance, National Chengchi University, Taipei 116, Taiwan; lifang@nccu.edu.tw

**Keywords:** appointments and schedules, online systems, outpatient clinics, hospital, Taiwan

## Abstract

Background: In the Internet era, many web-based appointment systems for hospitals have been established to replace traditional systems. Our study aimed to highlight the features of online appointment systems for hospitals in Taiwan, where patients can visit outpatient departments without a referral. Methods: All hospitals online appointment systems were surveyed in October 2018. Features of first-visit registrations were analyzed and stratified according to the hospitals’ accreditation levels. Results: Of the 417 hospitals, 59.7% (249) had public online appointment systems. For first-visit patients, only 199 hospitals offered the option of making appointments online from 7 to 98 (mean 38.9) days prior to the appointment itself. Before appointments, 68 (34.2%) hospitals recommended specialties for patients to choose according to their symptoms, and only 11 (5.5%) had a function for sending messages to doctors. After appointments, 176 (88.4%) provided links to real-time monitoring of outpatient service progress. Conclusions: More than half of the hospitals in Taiwan have public online appointment systems. However, most of these systems simply fulfill the function of registration, and rarely take the opportunity to improve efficiency by gathering information regarding patients’ medical history or reasons for making the appointment.

## 1. Introduction

### 1.1. Background of Online Registration Systems

Nowadays, there are many means by which to schedule a medical appointment. In the past, people used to make hospital appointments with schedulers in person or via telephone. However, these approaches may negatively influence patient satisfaction because they require verbal communication with real people who sometimes make mistakes, such as filling in the wrong appointment date or time, or sending the patient to the wrong health service provider [1]. Due to limited staffing and phone lines, appointments can only be made certain times, and waiting times for registering were often prolonged and inflexible [2].

Owing to the Internet’s rapid development, online medical services have become an important source of medical information for patients [3]. In many countries, web-based scheduling systems, with asynchronous and real-time modes, have been established [4], such as the “Choose and Book” appointment system of the National Health Service (NHS) in the United Kingdom, and the “web-based appointment systems (WAS)” in China [5,6]. With the assistance of such web-based appointment systems, patients seeking medical care can flexibly schedule an outpatient appointment and easily obtain real-time information [7]. Indeed, some studies have also demonstrated that online scheduling has positive impacts in terms of reducing non-attendance rates [5,8], decreasing staff labor [4], decreasing waiting times [4], and improving satisfaction rates [9].

### 1.2. Online Medical Records and Registration Systems in Taiwan

Taiwan has a flourishing Information Technology industry and a high Internet penetration rate. By 2017, the number of Internet users in Taiwan reached 18.8 million, accounting for 80% of the population [10]. Taiwan established its National Health Insurance system in 1995, which provides comprehensive healthcare coverage to all the nation’s citizens, including inpatient, outpatient, dental, and traditional Chinese medical services. The system covered 99.6% of Taiwan’s population, and had service contracts with about 93% of the country’s hospitals and clinics, as of 2017 [11]. In 2016, the average number of outpatient clinic visits per year for a patient was 15.4 [11]. Increasing numbers of health-related services have been developed to establish a better healthcare environment. Using the Internet, in 2013 the National Health Insurance Administration (NHIA) of the Ministry of Health and Welfare in Taiwan set up the “PharmaCloud System”, which enables physicians to search patients’ medical records to avoid adverse drug events [12]. The NHIA also established the “My Health Bank” service in September 2014, which enables patients to inspect their own medical information online, and narrows the information asymmetry between healthcare providers and patients [13].

Moreover, under Taiwan’s National Health Insurance system, patients can make appointments online with different specialists within hospitals [14], in person at hospital counters, and via telephone. Our previous study showed that 36.2% of hospitals in Taiwan offered an Android app for public use, and that 94.7% of these apps included an appointment booking function [15]. Another previous study showed that 51.1% of hospitals in Taiwan had an official Facebook fan page [16]. To completely survey the usage of online appointment systems, it is important to investigate how hospitals in Taiwan utilize Internet websites to facilitate registration, service patients, increase outpatient volumes, and even enhance public health.

This study aimed to provide an overview of the website-based online registration systems of all the hospitals in Taiwan. We focused on the information required for registration, as well as other features, of the relevant websites. The results of our study provide hospital managers with constructive insights regarding the establishment of online appointment systems.

## 2. Materials and Methods

### 2.1. Data Collection

All 417 hospitals in Taiwan, which had qualified for government-approved accreditation from 2013 to 2016, were surveyed. They received accreditation from the Taiwan Joint Commission on Hospital Accreditation, which is supervised by the Ministry of Health and Welfare, and were classified into three levels based on their healthcare quality, medical teaching ability, clinical capabilities, and hospital bed capacity. The three levels of classification were academic medical centers, regional hospitals, and local community hospitals.

The locations of the hospitals were categorized based on the urbanization stratification of Taiwan’s 368 townships developed by Taiwan’s National Health Research Institutes. All the townships were stratified into seven levels according to their demographic characteristics, industrialization, and medical resource distribution [17]. Of the 7 urbanization levels, we defined levels 1 and 2 as urban, levels 3 and 4 as suburban, and levels 5, 6, and 7 as rural. Two hospitals in the remote islands were stratified in the rural area.

We identified the official websites of all the hospitals in Taiwan by entering the given hospital’s name into a Google search engine, and then checked whether the official web pages had an online registration function or there were any links to their online registration system. We completed a first-time visitor online registrations, and recorded related data and characteristics for all the official online registration pages during the period from October 5^th^ to October 15^th^ of 2018. Any hospitals with links to online registration web pages that could not be opened were regarded as hospitals without an online registration systems.

### 2.2. Online Registration Page Characteristics Extraction

A Microsoft Excel worksheet was constructed to store the data extracted for all the online registration pages. The data was cross-sectional in nature. We accomplished first-visit online registrations to the cardiology departments of hospitals, because most hospitals had an outpatient clinic for their cardiology departments. As for those hospitals that had only one or two main divisions, such as divisions of obstetrics and gynecology or orthopedics, we chose to register with the given hospital’s main division.

Online registration can be divided into three stages. At the first stage, the features and characteristics that were provided by the web pages before a first-visit registration were inspected, which included division recommendations based on symptoms, direct links to web pages regarding doctors’ areas of expertise, information on the number of already registered patients, information on the maximum number of patients who could be treated, information on whether to show one’s full name at the given clinic, functions for sending messages to doctors, and English-language registration modes. At the second stage, we used our own identity card number and the related personal information required to accomplish first-visit online registrations, and we recorded all the items that we were definitely required to fill-in at the same time, whereas optional items were excluded. We also counted the number of days that were available for online registration. At the third stage, we inspected the features provided after registration, which included registration number display, the recommended arrival time, mobile phone message reminder functions, and outpatient service progress functions.

### 2.3. Statistical Analysis

In this study, descriptive statistics were compiled. We used the aforementioned Microsoft Excel worksheet to calculate the number of hospitals with an online registration system, the number of cases in which certain pieces of information were required to complete an online registration, and the total number of certain features provided by a given registration web page. The average with standard deviation, a bar chart, and one-way analysis of variance (ANOVA) test with a significance *p*-value cut-off of 0.05, were calculated and used to present the distribution of days available for making an online appointment before visiting, according to hospital type.

## 3. Results

### 3.1. Proportion of Hospitals with an Online Registration System

Of the 417 hospitals in Taiwan, 249 (59.7%) had a registration system on their website. All medical centers and most regional hospitals (98.8%) had an online registration system, while only about half the regional hospitals (47.3%) enabled patients to make an appointment online (Table 1). Stratified by urbanization level, a higher percentage of hospitals in urban areas (63.4%) had an online registration system compared to those in suburban (54.1%) and rural areas (51.9%).

### 3.2. Availability of Revisit and First-Visit Registration Functions

Of the 249 official websites with an online registration service, they all offered a revisit registration function, while 80.7% of hospital websites offered a first-time visit registration function. Among the three accreditation levels, all of the academic medical centers and 92.5% of the regional hospitals offered a first-visit registration function, while only 72.0% of local community hospitals offered the service (Table 2).

As shown in Table 2, an ID card number was most commonly required item for online registration, with nearly all hospitals requiring an ID card number to be filled in for first-time visit registration (98.5%). Birth year and date was the second most commonly required item for registration (88.6%). Almost all hospitals offered an online cancelation function (99.2%).

### 3.3. Average Time Available for Making an Online Appointment Before the Day of the Visit

Of the 201 official websites that allowed online first-visit registration, two did not indicate how long before a visit an appointment could be made, such that we could not collect further information; thus, they were excluded from further analysis. Among the 199 hospitals with a real-time online appointment mode, online registration could be completed on an average of 38.9 days before the day of the visit (Table 3). The average duration was shortest for academic medical centers, followed by that for regional hospitals, and local community hospitals (35.5, 38.0, and 40.2 days, respectively). However, there were no statistical differences between hospital categories (*p* = 0.610). Most hospitals (*n* = 106) offered an online appointment service within 21–30 days before the visit (Figure 1).

### 3.4. Features of Online Registration Systems Before and After First-Time Visits

We inspected the features that were available on the websites before and after a first-time visit registration. We found that, before registration, 34.1% of hospital registration pages provided division recommendations based on patient symptoms, while 32.7% of the pages offered links to webpages regarding doctors’ areas of expertise (Table 4). In addition, 41.7% of hospitals displayed the number of already registered patients on the page. As for the features made available after registration was accomplished, we found that nearly all (99.5%) hospitals displayed the relevant registration number, while 19.6% of the web pages displayed a suggested arrival time. Moreover, 88.4% of the official hospital registration pages provided real-time data regarding outpatient service progress.

## 4. Discussion

### 4.1. Distribution of Online Appointment Systems

In our study, we found that the proportion of hospitals offering appointments on their websites was much higher among academic medical centers and regional hospitals, than local community hospitals. This finding might be related to the different outpatient volumes and incomes at different hospital levels. Academic medical centers in Taiwan have higher outpatient department service volumes than other hospitals. According to statistics from a 2017 annual report of Taiwan’s Ministry of Health and Welfare, the average number of outpatients seen per day is 4977.3 for an academic medical center, 2168.7 for a regional hospital, and only 369.3 for a local community hospital [18]. Academic medical centers are often located in urban areas and receive higher levels of financial support and medical resources than other hospitals. Therefore, they are the most popular choice among patients in Taiwan even though they generally charge more for the services they provide. In contrast, local community hospitals are smaller in scale, receive lower levels of funding, and serve fewer patients. The number of local community hospitals decreased from 451 in 1997, to 333 in 2009 due to the difficulties faced in managing such hospitals. A probable reason for this decline was that patients cared more about the higher quality of medical services provided by academic medical centers than about the lower fees charged by local clinics [19]. Therefore, local community hospitals were less willing to provide online appointment systems due to the high costs of setting up and maintaining them. However, we also found that some local community hospitals had the same appointment systems as academic medical centers because those local hospitals are branches of larger hospital systems, and thus shared the same web systems as other hospitals (such as academic medical centers). Otherwise, it is notable that there were few differences among hospitals with different levels of urbanization in terms of offering the option to make appointments online, suggesting that the urban–rural gap in Internet development is not apparent in this respect.

### 4.2. First-Time Online Appointments

In this study, we found that 249 hospitals in Taiwan offered revisit appointment registration on their websites, and that 201 (80.7%) offered first-visit registration. Almost all hospitals required ID card or passport numbers for registration. An ID card number is a unique number used to identify someone, as the number used on a given card will not be repeated on another. The prospective patient’s birth year and date were also commonly required items. For security reasons, some hospitals needed to key in a random verification code, which can probably avoid the problem of mass appointments being scheduled by bots. In addition, other items were also commonly required for first-visit appointment registrations, such as the patient’s name, address, and telephone number.

Some hospitals provided forms to acquire additional medical information in advance, such as the given patient’s past medical history, family history, allergy history, and social history, including alcohol drinking, betel nut chewing, and cigarette smoking. By filling out these forms in advance, patients saved time on the day they arrived at the hospital. However, these forms were all optional for online registration; patients could also choose to answer the relevant questions by filling in questionnaires after arriving at the given hospital. Meanwhile, almost all (99.2%) hospitals provided a function for cancelling a previously registered appointment. The two hospitals without such a cancelation function offered only asynchronous appointment registration, meaning that an appointment could only be made by sending an email to schedulers at the given hospital, who would then schedule the appointment manually. This approach was a choice made by these two smaller-scale hospitals to reduce costs on maintaining their online appointment systems. All other hospital appointment systems, however, allowed for real-time registration.

### 4.3. Features of Online Appointment Systems

With a convenient appointment system, hospitals have better engagement with patients and provide care to more patients [20]. However, an appointment system can also be a source of dissatisfaction for both patients and health providers if it is not well designed [21]. In this study, we found that 68 (34.1%) hospitals offered division recommendations based on patients’ symptoms, while 65 (32.7%) hospitals provided direct links to webpages regarding doctors’ areas of expertise from their appointment webpages. These results showed that some patients learn which specialists to visit from the appointment systems but not from their family physicians. That might imply that the family physician’s trial plan in Taiwan should be promoted further, even though there have already been some benefits from the plan [22]. Meanwhile, only 45 (23.0%) hospitals offered English-language registration, which reflects the fact that the internationalization of Taiwan thus far is insufficient, and that future efforts are required in order to serve more patients from different countries [23]. Due to privacy concerns, 24 (12.1%) hospitals offered patients the choice of whether or not to show their full name at a clinic. Only 11 (5.5%) hospitals offered the function of sending messages directly to doctors. By gathering information prior to medical visits, healthcare providers can be more prepared, and unnecessary waiting times and misunderstandings can be reduced.

After an appointment was made, 39 (19.6%) hospitals displayed a recommended arrival time, while 176 (88.4%) hospitals provided real-time data regarding outpatient service progress online. These functions are aimed at reducing patients’ waiting times at the hospital itself. The registration systems we reviewed in this study did not offer the ability to pick a specific time for an appointment, but they often indicated the given patient’s place in line after the patient chose a time slot. There was no limit placed on the maximum number of appointments per day for some physicians, which could result in prolonged waiting times and crowded waiting rooms. In this context, the question of how to reduce waiting times has become an important issue. Patients who fail to show up for an appointment are a significant cause of wasted clinical resources [5]. To reduce no-show rates, some hospitals have announced that if a patient fails to show up for appointments a certain number of times, their ID card number will be put onto a “blacklist”, meaning that they will be forbidden from receiving services at the hospital for a period of time as punishment. Previous studies have revealed that short message service reminders can increase the likelihood that patients will attend clinic appointments [24,25]. However, only 6 (3.0%) hospitals reviewed in this study provided phone text messages as reminders for patients. To create a more convenient healthcare environment, adding these features within appointment systems should be considered.

### 4.4. Online Appointment Systems in Taiwan and other Countries

In spite of Taiwan’s high Internet penetration rate, only 59.7% of hospitals in Taiwan offer online appointment services, and the features and quality levels of these services are inconsistent. A previous study reported that only~15% of public hospitals, and 18% of private hospitals in Italy allowed appointments to be made online in 2008–2009 [26]. Another study revealed that ~7% of primary care practices in Canada, and 30% in the United States offered web-based appointment services in 2012 [27]. However, while the prevalence of web-based appointment systems was higher in Taiwan than in some developed countries, most of the online appointment systems in Taiwan simply focus on registration and providing a waiting number, without gathering further medical information from patients or allowing for requests to physicians before the day of an appointment.

Another study analyzed how hospitals in Spain used webpages and social networks with three different dimensions: presence, use, and impact. The results showed that 69.9% of hospitals had an institutional webpage, and 34.2% used at least one of the social networks (Facebook, Twitter, and YouTube) in 2015 [28]. The Internet presence of Spanish hospitals is high, but their presence on the social networks is not as high compared to that of hospitals in the United States [29]. Our previous study showed that only 51.1% of hospitals in Taiwan had an official Facebook fan page in spite of high Internet penetration rate in Taiwan [16]. These study results showed that social media does not seem to be the main method for citizens to approach medical resources in some countries, especially in making medical appointments. More attention should be paid to incorporate social media into healthcare to create a more efficient environment.

### 4.5. Limitations

There were some limitations to this study. First, it was a cross-sectional study in nature. We recorded the features for revisit and first-visit registrations in October 2018, such as the items required for registration, the days available for registration, and other characteristics, but these features may change over time. Second, we did not have a valid ID card number with which to complete revisit registrations for all the hospitals; thus, we did not analyze the information that would only be shown after we entered a valid revisit ID card number. As such, our study results were mainly focused on the features of first-time visit registrations, even though revisit and first-time registrations used the same appointment systems. Finally, this study only reviewed online appointment systems that were publicly available in Taiwan, and so its results are not representative of all the systems available in markets around the world.

## 5. Conclusions

More than half the hospitals in Taiwan have public online appointment systems. However, most systems simply fulfill the function of registration while rarely taking the opportunity to gather additional information, such as the related medical history or reasons for consultation. Further efforts should thus be devoted to strengthening the functions of these online appointment systems so that the efficiency of consultations can be improved. Some methodologies for information extraction and retrieval on unstructured medical records, such as a lexicon-grammar based methodology [30], should be taken into consideration to enhance appointment process efficiency.

## Figures and Tables

**Figure 1 ijerph-16-00171-f001:**
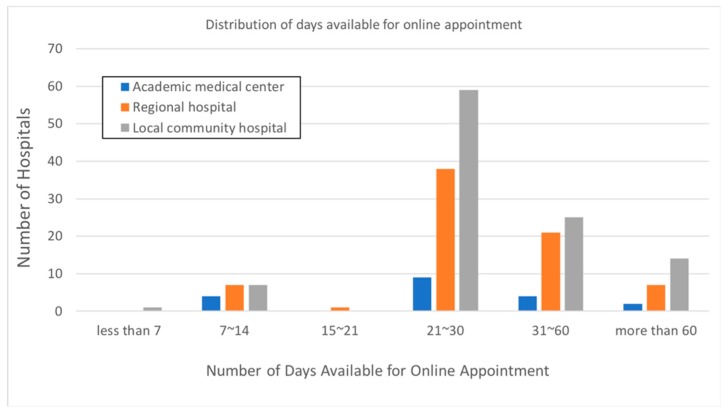
Distribution of days available for making an online appointment, stratified by hospital accreditation type.

**Table 1 ijerph-16-00171-t001:** Proportion of hospitals with an online registration system among all hospitals in Taiwan, stratified by accreditation type and urbanization level.

	Academic Medical Center	Regional Hospital	Local Community Hospital	Total
Urban	100% (19/19) ^a^	98.2% (55/56)	48.9% (89/182)	63.4% (163/257)
Suburban	N/A ^b^	100% (24/24)	44.0% (48/109)	54.1% (72/133)
Rural	N/A	100% (1/1)	50.0% (13/26)	51.9% (14/27)
Total	100% (19/19)	98.8% (80/81)	47.3% (150/317)	59.7% (249/417)

^a^ Values are the number of such hospitals with an online registration system/the number of all such hospitals. ^b^ N/A: not applicable.

**Table 2 ijerph-16-00171-t002:** The number of hospitals offering a first-visit registration function, and requiring a given item for registration, as well as the number of hospitals offering a cancelation function, stratified by accreditation type.

	Academic Medical Center (*n* = 19)	Regional Hospital (*n* = 80)	Local Community Hospital (*n* = 150)	Total (*n* = 249)
**First-visit registration**	19	74	108	201
ID card number	19 (100%)	73 (98.6%)	106 (98.1%)	198 (98.5%)
Birth year and date	17 (89.5%)	65 (87.8%)	96 (88.9%)	178 (88.6%)
Name	12 (63.2%)	52 (70.3%)	81 (75.0%)	145 (72.1%)
Address	11 (57.9%)	28 (37.8%)	23 (21.3%)	62 (30.8%)
Telephone number	14 (73.7%)	49 (66.2%)	68 (63.0%)	131 (65.2%)
Verification code	9 (47.4%)	35 (43.8%)	64 (59.3%)	108 (53.7%)
**Cancelation**	19	80	148	247

**Table 3 ijerph-16-00171-t003:** Average, maximum, minimum, and median number of days available for making a first-visit appointment online before the day of the visit, stratified by hospital accreditation type.

	Academic Medical Center	Regional Hospital	Local Community Hospital	Total
	(*n* = 19)	(*n* = 74)	(*n* = 106)	(*n* = 199)
Average (M ± SD)	35.5 ± 21.4	38.0 ± 20.3	40.2 ± 22.1	38.9 ± 21.5
Max	90	98	98	98
Min	14	14	7	7
Median	30	30	30	30

Abbreviations: M = mean; SD = standard deviation.

**Table 4 ijerph-16-00171-t004:** Features of real-time online registration system websites before and after a first-time visit, stratified by hospital accreditation type.

	Academic Medical Center (*n* = 19)	Regional Hospital (*n* = 74)	Local Community Hospital (*n* = 108)	Total (*n* = 199)
**Before registration**				
Recommend specialties by symptoms	14 (73.7%)	21 (28.4%)	33 (30.6%)	68 (34.1%)
Links to webpages regarding doctors’ areas of expertise	11 (57.9%)	24 (32.4%)	30 (27.8%)	65 (32.7%)
Show number of already registered patients	6 (31.6%)	38 (51.4%)	39 (36.1%)	83 (41.7%)
Indicate whether to show full name at clinic	8 (42.1%)	10 (13.5%)	6 (5.6%)	24 (12.1%)
Function for sending messages to doctors	0 (0%)	4 (5.4%)	7 (6.5%)	11 (5.5%)
English-language appointment systems	11 (57.9%)	19 (25.7%)	15 (13.9%)	45 (23.0%)
**After registration**				
Display registration number	19 (100%)	73 (98.6%)	106 (98.1%)	198 (99.5%)
Display a suggested arrival time	6 (31.6%)	18 (24.3%)	15 (13.9%)	39 (19.6%)
Mobile phone text message reminders	0 (0%)	2 (2.7%)	4 (3.7%)	6 (3.0%)
Real-time data regarding outpatient service progress	19 (100%)	71 (95.9%)	86 (79.6%)	176 (88.4%)

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
