# Peer review of "Features of Online Hospital Appointment Systems in Taiwan: A Nationwide Survey"

_ijerph, 2019, doi:10.3390/ijerph16020171_

Round 1

Reviewer 1 Report

IJERPH-402560: Features of Online Hospital Appointment Systems in Taiwan: a Nationwide Survey

The paper has many issues for its consideration as a journal article. As against the title, features of online appointment system seek quite vague and incomplete. Similarly, abstract notes that all the hospitals were surveyed while just checking their websites would not lead to satisfactory data as contained in the methods section. The introduction does not provide any insight on how this work would contribute to the science. One of my main observations is the lack of proper scientific vigor while many simple aspects of online registration may end up consuming journal pages. Authors note that following aspects need to be studied (line 77 onward):

‘To completely survey the usage of online appointment systems, it is important to investigate how hospitals in Taiwan utilize Internet websites to facilitate registration, service patients, increase outpatient volumes, and even enhance public health.’ But then they just confine themselves to website-based online registration systems. To come up with effectiveness of such interventions, in my view, a survey of some of the end-users could effectively lead to interesting insights which seems missing here. Discussion and conclusions are either vague or provide limited information.

Author Response

Dear Sir

Thank you very much for reviewing our manuscript for publication in the “International Journal of Environmental Research and Public Health”.

As to your comments, we would like to answer as follows:

Comments: 

The paper has many issues for its consideration as a journal article. As against the title, features of online appointment system seek quite vague and incomplete. Similarly, abstract notes that all the hospitals were surveyed while just checking their websites would not lead to satisfactory data as contained in the methods section. The introduction does not provide any insight on how this work would contribute to the science. One of my main observations is the lack of proper scientific vigor while many simple aspects of online registration may end up consuming journal pages. Authors note that following aspects need to be studied (line 77 onward):

‘To completely survey the usage of online appointment systems, it is important to investigate how hospitals in Taiwan utilize Internet websites to facilitate registration, service patients, increase outpatient volumes, and even enhance public health.’ But then they just confine themselves to website-based online registration systems. To come up with effectiveness of such interventions, in my view, a survey of some of the end-users could effectively lead to interesting insights which seems missing here. Discussion and conclusions are either vague or provide limited information.

Response: 

Thanks for the reviewer’s valuable comment. Our study is a descriptive, cross-sectional study, aimed to provide an overview of the website-based online registration systems of all the hospitals in Taiwan. Taiwan has a unique healthcare system that patients can visit any outpatient departments without a referral. Among all the hospitals with online appointment system, appointments can be made through their official websites, and there is no one individual platform to make appointments to different hospitals at a time. Thus, checking official websites of all hospitals is sufficient for surveying the features of online hospital appointment systems. The results of our study might facilitate international comparison, and provide hospital managers with constructive insights regarding the establishment of online appointment systems. However, the level of citizens using online appointment system is difficult to quantify. Information such as service volume and the proportion of patients using online system as a source of registration were not publicly available. Thus, we focused on the information required for registration, as well as other features of the system.Further efforts should be devoted to strengthening the functions of these online appointment systems, and considering more end users’ feedbacks and experiences, so that the efficiency of consultations can be improved.

Thanks again for the reviewer’s constructive comments. 

Reviewer 2 Report

The manuscript presents a descriptive study on the characteristics of on-line appointment systems in Hopsitals in Taiwan. I enjoyed reading the manuscript, which is very-well structured and clear.

English writing and some sentences could be improved, so I suggest authors to accurately revise it.

I have no major comments to the manuscript, but I have minor suggestions:

1) Add % in table 4 so it is easier to understand the differences between academic and regional centers. With absolute values is hard to understand the differences.

2)I would suggest authors to run easy statistical tests to proof the homogenity or heterogenity of numeric variables and categorical features. This would be as simple as runing a one-way ANOVA test for categorical distributions to demonstrate statistical differences between hospital categories.

3) Authors could also discuss about the position of taiwan hospitals in social media as a way to build awareness of such online appointment systems. You can use this paper as a reference for discussion: https://www.jmir.org/2017/5/e181/

Author Response

Dear Sir

Thank you very much for reviewing our manuscript for publication in the “International Journal of Environmental Research and Public Health”.

As to your comments, we would like to answer as follows:

Comments:

1) Add % in table 4 so it is easier to understand the differences between academic and regional centers. With absolute values is hard to understand the differences.

Response 1:Thanks for the reviewer’s valuable comment. We have revised table 4 to make it easier to understand the differences between academic and regional hospitals.

2) I would suggest authors to run easy statistical tests to proof the homogenity or heterogenity of numeric variables and categorical features. This would be as simple as runing a one-way ANOVA test for categorical distributions to demonstrate statistical differences between hospital categories.

Response 2:Thanks for the reviewer’s valuable comment. We have run a one-way ANOVA test on the number of days available for making a first-visit appointment online before the day of visit. The result showed that there were no statistical differences between hospital categories (p=0.610). 

We have revised the sentences in the Materials and Methods section as follows: “In this study, descriptive statistics was compiled. We used the aforementioned Microsoft Excel worksheet to calculate the number of hospitals with an online registration system, the number of cases in which certain pieces of information were required to complete an online registration, and the total number of certain features provided by a given registration web page. Averages with standard deviations and a bar chart, and a one-way analysis of variance (ANOVA) test with a significance p-value cut-off of 0.05 were also calculated and prepared to present the distribution of the numbers of days available for making an online appointment before visiting, according to hospital type.” 

We have also revised the sentences in the Results section as follows: “Of all the 201 official websites that allowed for online first-visit registration, two of them did not indicate how long before a visit an appointment could be made, such that we could not collect further information; thus, they were excluded from further analysis. Among the 199 hospitals with a real-time online appointment mode, online registration could be completed an average of 38.9 days before the day of the visit (Table 3). The average duration was shortest for academic medical centers, followed by that for regional hospitals and local community hospitals (35.5, 38.0, and 40.2 days, respectively). However, there were no statistical differences between hospital categories (p=0.610). Most of the hospitals (n=106) offered an online appointment service within the range of 21 to 30 days before the visit (Figure 1).”. 

3) Authors could also discuss about the position of taiwan hospitals in social media as a way to build awareness of such online appointment systems. You can use this paper as a reference for discussion: https://www.jmir.org/2017/5/e181/

Response 3:Thanks for the reviewer’s valuable comment. In the recent decade, social media have become a part of life for many people in the world. For example, of approximately 23 million inhabitants, there are 18 million Facebook active accounts in Taiwan. However, in spite of the popularity of Facebook among the general population, only 51.1% of the hospitals have a Facebook page according to our previous study. Hospitals in Taiwan did not seem to make good use of social media.

We have revised the sentences in the Discussion section as follows: “Another previous study had analyzed how hospital in Spain used webpage and social network with 3 different dimensions: presence, use, and impact. The results showed that 69.9% of hospitals had an institutional webpage and 34.2% had used at least one of the social networks (Facebook, Twitter, and YouTube) in 2015 [28]. The Internet presence of Spanish hospitals is high, but their presence on the social networks is not as high compared to that of hospitals in the United States [29]. Another our previous study showed that only 51.1% of hospitals in Taiwan had an official Facebook fan page in spite of high Internet penetration rate in Taiwan [16]. These study results showed that social media still does not seem to be the main method for citizensto approach medical resources in some countries, especially inmaking medical appointments. More attention should be paid to incorporate social media into healthcare to create a more efficient environment.” 

Accordingly, we had revised the manuscript. Thanks again for your constructive comments.

Reviewer 3 Report

The authors provide a survey about Online Hospital Appointment Systems in Taiwan by examining in deep the features and characteristic of entire appointment process.

The methodology is interesting but some points should be better explained. In the section 2 a description of the main activities/phases of the appointment process and how they are performed could be useful for better understanding the examined problem. A linguistic revision is necessary to correct some mistakes in the paper (i.e.the caption of the Figure 1 is Nunber of...).

Furthermore, my suggestion is to analyze also more recent approaches (2015,2016,2017) about the examined topics. Finally, in the conclusion section I suggest to cite the following paper for investigating how ontology can easily support the appointment process:

1) A lexicon-grammar based methodology for ontology population for e-health applications. In Complex, Intelligent, and Software Intensive Systems (CISIS), 2015 Ninth International Conference on (pp. 521-526). IEEE.

Author Response

Dear Sir

Thank you very much for reviewing our manuscript for publication in the “International Journal of Environmental Research and Public Health”.

As to your comments, we would like to answer as follows:

Comments:

A) The authors provide a survey about Online Hospital Appointment Systems in Taiwan by examining in deep the features and characteristic of entire appointment process.

The methodology is interesting but some points should be better explained. In the section 2 a description of the main activities/phases of the appointment process and how they are performed could be useful for better understanding the examined problem. A linguistic revision is necessary to correct some mistakes in the paper (i.e. the caption of the Figure 1 is Nunber of...).

Response A:Thanks for the reviewer’s valuable comment. We have revised the sentences in the Materials and Methods section for better understanding the examined problem as follows: “A Microsoft Excel worksheet was constructed to store the data extracted for all the online registration pages. The data was cross-sectional in nature. We accomplished first-visit online registrations to the cardiology departments of the hospitals, because most of the hospitals had an outpatient clinic for their cardiology departments. As for those hospitals that had only one or two main divisions, such as divisions of obstetrics & gynecology or orthopedics, we chose to register with the given hospital’s main division.

Online registration can be divided into three stages. At the first stage, the features and characteristics that were provided by the web pages before a first-visit registration were inspected, which included division recommendations based on symptoms, direct links to webpages regarding doctors’ areas of expertise, information on the number of already registered patients, information on the maximum number of patients who could be treated, information on whether to show one’s full name at the given clinic, functions for sending messages to doctors, and English-language registration modes. At the second stage, we used our own identity card number and the related personal information required to accomplish first-visit online registrations, and we recorded all the items that we were definitely required to fill-in at the same time, whereas optional items were excluded. We also counted the number of days that were available for online registration. At the third stage, we inspected the features provided after registration, which included the display of the registration number, the recommended arrival time, mobile phone message reminder functions, and outpatient service progress functions.”

We have also corrected linguistic mistakes in the paper.

B) Furthermore, my suggestion is to analyze also more recent approaches (2015,2016,2017) about the examined topics. Finally, in the conclusion section I suggest to cite the following paper for investigating how ontology can easily support the appointment process:

1) A lexicon-grammar based methodology for ontology population for e-health applications. In Complex, Intelligent, and Software Intensive Systems (CISIS), 2015 Ninth International Conference on (pp. 521-526). IEEE.

Response B:Thanks for the reviewer’s valuable comment. In fact, we have searched PubMed by using related keywords. We have downloaded more than 50 articles. After carefully reviewing the articles, we chose the most important ones as our references. However, the articles analyzing online appointment system are relatively few. 

We have cited the suggested reference to enrich our manuscript, and revised the sentences in the Conclusionsection as follows: “More than one half of the hospitals in Taiwan have online appointment systems for the public. However, most of these systems simply fulfil the function of registration while rarely taking the opportunity to gather additional information such as the related medical history or reasons for consultation. Further efforts should thus be devoted to strengthening the functions of these online appointment systems so that the efficiency of consultations can be improved. Some methodology for information extraction and retrieval on unstructured medical records such as a lexicon-grammar based methodology [30] should be taken into consideration to enhance appointment process efficiency.”

Accordingly, we had revised the manuscript. Thanks again for your constructive comments.

Round 2

Reviewer 1 Report

The paper can be considered for publication at this point after convinced by the arguments of the author.